# Influences of Urban Discharges and Urban Heat Effects on Stream Temperature

Anja Svane Kolath * and Sara Egemose

Department of Biology, University of Southern Denmark, DK-5230 Odense, Denmark
* Correspondence: anjasvane@biology.sdu.dk

**Abstract:** Urban areas with dark and impermeable surfaces are known to have a heating effect on air and still water compared to surrounding areas, called the urban heat island effect (UHI). UHI and stormwater discharges' collective impact on stream temperature, especially regarding seasonal changes, is a less-studied field. In this study, the temperature effect of the urban village Aarslev on Stream Vindinge in Southern Denmark was examined. Loggers (ID A–L) were placed in Stream Vindinge in 2020–2021, measuring temperature (°C) and pressure (kPa). Outlets were analyzed with respect to origin: Direct stormwater outlets (rain ÷ basin), stormwater delayed by ponds (rain + basin), common overflow, and common sewage from WWTP. Data showed the stream temperature rise through Aarslev village in all months (except March) with 0.3–1.9 °C, most notably in the summer months. A one-way ANOVA confirmed that the upstream station A and downstream station K were significantly different ($p$-values < 0.001). No significant difference in temperatures between the different outlet types was found. An increase in stream temperature was observed in response to rain events, followed by a temperature decrease. This was assumed to be a "first heat flush". This was speculated to mean less optimal conditions for trout and sensitive macroinvertebrates not because of heat shock, but rather to lower $O_2$ concentrations and higher mineralization. River and lake temperatures are projected to increase, and this effect might become more pronounced. A decrease in stream temperature was observed after the village (station L). Therefore, it was concluded that the rise in temperature through the village was due to UHI.

**Keywords:** urban heat island effect; stream temperature; urban environment; yearly variations; different outlet types; heat shock; trout; macroinvertebrates



## 1. Introduction

Climate change has, in many parts of the world, become more easily observable in recent years, with most of the world experiencing changes in heat and precipitation patterns [1–4]. Earth's air temperature has, on average, increased by 0.6 °C in the last 100 years. In Denmark, the temperature has increased by 1.4 °C in the last 150 years [5].

Urban landscapes are known to have a further heating effect on the air and water temperature compared to surrounding areas [6–8]. Temperature differences of up to 10 °C have been noted between cities and the surrounding areas [9]. The expansion of urbanization includes the implementation of paved surfaces (roofs, asphalt roads, etc.) which absorb, retain, and conduct heat more efficiently than natural surfaces (grass, dirt, earth, etc.) [10,11]. This phenomenon is known as the urban heat island (UHI). The absorbed heat can be dissipated to stormwater runoff during rain events [10,11] and lead to lakes and streams. Previous studies have shown that an increase in impermeable area coverage (%) is positively linked to an increase in stream temperature in relation to rain events [12–15]. Different studies have yielded different results, however with every 1% increase in impermeable catchment area, a temperature increase of 0.09–0.37 °C in the recipient streams has been observed [13,14].

Precipitation in the northern regions is projected to increase as a consequence of climate change [16]. Precipitation has increased in Denmark with an average of 11 mm/decade in the last 150 years [5] and is projected to increase with 50–100 mm towards year 2100, depending on scenario [17]. In northern Europe the seasonal precipitation patterns are also expected to change, leading to more precipitation in spring, autumn, and winter because of the overall increase in precipitation and because more will be in the form of rain instead of snow [18,19]. Overall runoff is expected to increase by 10–30% in 2071–2100 [20]. Urban runoff is often redirected through stormwater control measures or green infrastructures, such as wet stormwater ponds or rain outlets before discharge to streams. However, stormwater control measures are rarely designed with thermal regulation in mind. Wet stormwater ponds are designed to look and function like natural small lakes and rely on natural processes for cleaning and delaying stormwater before discharge to stream recipient [21–23]. Examinations of temperature differences in wet stormwater ponds have indicated discharge of water with elevated temperatures, with large variations. Herb et al. [24] estimated that wet stormwater detention ponds outflow temperature would, on average, be 1.2 °C warmer than inflow, with large variations between events. Other studies have shown temperature increases of up to 9.0 °C [25], while others found stormwater pond discharge to only result in short pulses of higher stream temperature to a maximum of 1.5 °C [26].

The combination of higher temperatures and increased precipitation will likely affect the receiving freshwater ecosystems. Receiving freshwater ecosystems depends on having a predictable yearly circle with small or natural fluctuations to thrive. Many factors can impact this ecosystem, such as nutrient load, dissolved oxygen content, and the temperature. A rise in temperature can be problematic because this leads to an increase in bacterial activity and $O_2$ demand, leading to oxygen-depleted areas [27–31]. A rise in stream temperature, both continuously and as a pulse, may also directly impact the animals that live in streams, as temperature influences growth, metabolism, reproduction, and ultimately survival, if the highest thermal limit is exceeded [29,30]. Animals that require a high oxygen content and low temperatures will likely suffer in the future with the predicted temperature increases caused by climate changes. Trout, salmons, stoneflies, caddisflies, and mayflies are some species that would be expected to be strongly affected, as they are cold-water species with a high $O_2$ demand [27]. Brown trout (*Salmo trutta*) have a critical thermal maxima (CTM) of 30 °C [32], but lower for alevins (20–24 °C) and for parr and smolt (22–30 °C) [33–35]. Examinations on a common Danish Plecoptera species (*Leuctra hippopus*) have shown that eggs will hatch in a temperature interval of 2–24°C, but nymphs have a higher survival rate in a temperature interval of 12–20 °C [36]. Studies of other sensitive common species include the mayfly family *Baetidae* being shown to have an upper thermal tolerance (UTT) of 20.1 °C [37] or the caddisfly family *Limnephilidae* having an UTT of 26.1 °C [37,38]. Local variations and changes in stream temperatures are just one of many factors that affect the streams ecosystem, but it is a less-studied field compared to the effects of geomorphology, hydrology, and nutrients [8,39]. Some studies have been done on the small-scale variations in stream temperatures, but it has often been done in connection to salmon and trout fisheries and the effect of dams and loss of forested buffer zones [40–42]. The general effect of stream temperatures as an abiotic factor has been the subject of studies for years [14,27–31,43–47]. However, the study of stream temperature affected by UHI is a less-studied field [15]. For UHI effects the main focus has been on air- and surface temperatures, but the presence of a hydrological urban heat island (HUIH) has been speculated [15]. The purpose of this study was to help fill this knowledge gap focusing on both UHI and runoff discharge both directly and delayed through stormwater ponds. In this study, a stretch of Stream Vindinge that passes through Aarslev village in the region of Southern Denmark was monitored using stationary temperature and pressure loggers for 2 years.

## 2. Materials and Methods

### 2.1. Study Area

This study was performed in Stream Vindinge on the stretch that runs through the village of Aarslev in Southern Denmark. Denmark has a semiarid climate with a yearly precipitation of 782 mm and an average yearly temperature of 8.7 °C [48,49]. Stream Vindinge is roughly 30 km long and originates southwest of Aarslev [50]. The examined stretch of Vindinge stream has a width of roughly 199 cm and a depth ranging from 3–33 cm, with an average depth of 21 cm. Aarslev village has both separate and combined sewage pipes. Most of the village has separate pipes for stormwater, which is led into either a pond or Stream Vindinge. The combined sewer network receives stormwater from small sections of the village and mainly receives the wastewater, which is led to the treatment plant.

### 2.2. Loggers and Placement

Twelve loggers (A–L) were placed in Stream Vindinge upstream, through and downstream of Aarslev village, comprising a stretch of stream of roughly 5.2 km (Figure 1). The loggers were placed downstream of different outlet types to gain an impression of the changes in temperature and depth. These outlet types comprise rainwater outlets with or without a stormwater pond (Rain ± basin), common overflows, and an outlet from the wastewater treatment plant (Common (WWTP)). The loggers in the village were placed 10–300 m downstream of the nearest outlet, while station K and L were placed after the village (Table 1).

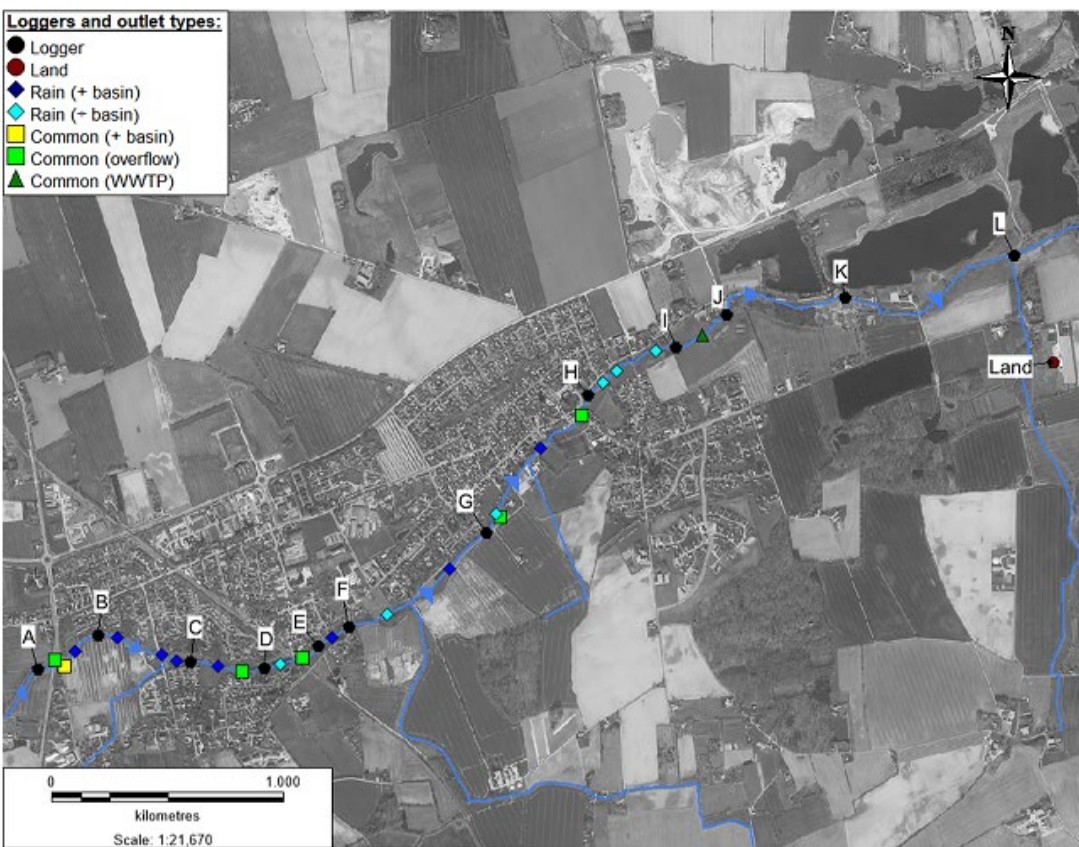

**Figure 1.** Logger placement and outlet type (Rain (÷basin), Rain(+basin), Common (overflow), Common (WWTP) and land), in Stream Vindinge through the city of Aarslev.

At station A, B, C, D, F, G, K HOBO Water Level Data Loggers that measure pressure [kPa] and temperature [°C] were placed. Loggers at station E, H, I, and L were HOBO Pendant MX2201 Temperature data loggers that only measure temperature [°C]. At the "Land" station a Water Level logger was placed on land as a reference for atmospheric

pressure, which was used to calculate the water depth in the stream. A HOBO Rain Gauge Data Logger was also placed at the "Land" station. The rain gauge measured temperature [°C] and rainfall [mm]. The stations were all inspected, loggers were checked for damages, and data were collected once a month. This article comprises data from 27 February 2020 to 31 August 2021. Logger A was lost in the period 8 August–25 November 2020. Station B was occasionally used instead of station A to analyze results. All the loggers collected data every 10 min and attached chains were used to weigh down the loggers to the stream bed (Figure 2). A cord was further used to attach the loggers to nearby trees, poles, or rails to avoid relocation.

**Table 1.** Information about the 12 loggers placed in Stream Vindinge. Including Logger ID (A–L), coordinates (EUREF89_zone32), the nearest upstream outlet type of each logger divided in Rain(÷Basin), Rain(+Basin), Common (Overflow and WWTP), and distance to nearest upstream outlet [m].

| Logger ID | EUREF89_zone32 | Upstream Outlet Type | Distance from Outlet [m] |
|---|---|---|---|
| A | 592,146; 6,129,101 | Rain (÷Basin) | 10 |
| B | 592,416; 6,129,257 | Rain (+Basin) | 130 |
| C | 592,800; 6,129,167 | Rain (+Basin) | 10 |
| D | 593,140; 6,129,119 | Common (Overflow) | 100 |
| E | 593,316; 6,129,160 | Common (Overflow) | 10 |
| F | 593,503; 6,129,290 | Rain (+Basin) | 20 |
| G | 594,105; 6,129,684 | Rain (+Basin) | 150 |
| H | 594,534; 6,130,281 | Common (Overflow) | 100 |
| I | 594,906; 6,130,510 | Rain (÷Basin) | 300 |
| J | 595,042; 6,130,559 | Common (WWTP) | 25 |
| K | 595,650; 6,130,724 | Common (WWTP) | 725 |
| L | 596,396; 6,130,905 | Common (WWTP) | 1630 |

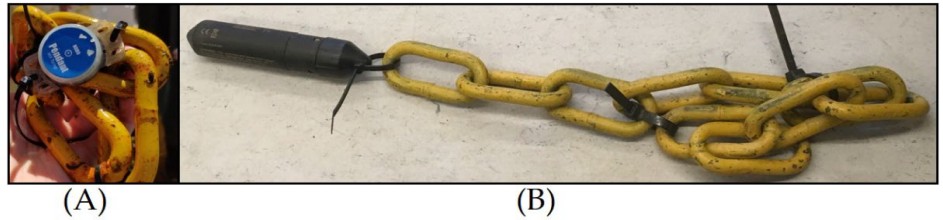

(A)  (B)

**Figure 2.** (**A**) HOBO Pendant MX2201 [°C] and (**B**) HOBO Water Level Data logger [kPa & °C].

### 2.3. Statistical Analysis

To analyze significant differences in measured temperatures between the stations, outlet types, and in different seasons, a one-way ANOVA was utilized (Table 2). Linear regressions were utilized to test the possible correlations between temperature and water depths and rain patterns. Statistical analyses were performed in Sigmaplot 13.0. For all tests, a significance level of $\alpha = 0.05$ was chosen.

**Table 2.** One-way ANOVA tests on ranks between the upstream station A and downstream station K for summer 2020 and 2021 and winter 2020. Including number of data points (N), diff of ranks, *p*-value, and result of test $p < 0.050$ (yes/no).

| One Way ANOVA on Ranks (Tukey Test) | | | | | |
|---|---|---|---|---|---|
| **Season** | **Stations** | **N** | **Diff of Ranks** | ***p*** | ***p* < 0.050** |
| Summer 2020 | A vs. K | 8760 | 14168388 | <0.001 | yes |
| Winter 2020 | A vs. K | 8926 | 20696869 | <0.001 | yes |
| Summer 2021 | A vs. K | 8768 | 7959435 | <0.001 | yes |

## 3. Results

### 3.1. Seasonal Stream Variations in Temperature and Water Depth

Through the continuous measurements of temperature and water depth variations in Stream Vindinge, some interesting trends became apparent. In Table 3, the monthly average temperatures (±SD) for 2020–2021 are listed with the individual stations indicated (station A to L). The temperature over the years ranged from 2.1 °C in February at station A to 17.5 °C at station K in August. The temperature increased with distance travelled through the village in almost all the months, with only a few variations in the fall. The lowest temperatures in each month were generally registered at station A (from 2.1 °C in February to 15.4 °C in July), while the highest temperatures were measured at station K and L (from 2.9 °C in February to 17.5 °C in July). Therefore, a more than 2 °C increase was observed through the village in the summer period. A few variations in this trend were seen in the fall, where station A in September had an average of 14.2 °C compared to an average between 13.6–13.8 °C for station B–F. In November, station A and D had the lowest temperatures (4.6 °C and 6.9 °C, respectively) compared to 7.8–8.0 °C for stations B, C, E, F, H, I, and an average temperature of 8.4–8.9 °C for station J–L. Focusing on only May–June in the summer and the winter of 2020 and 2021 for station A and station K (Figure 3), an interesting trend became clear: generally, the downstream station K had both larger temperature variations, but also higher temperatures in general compared to station A. The variations of station K are more pronounced in the summer period compared to the winter, but in both cases the air temperature seems to influence the water temperature of both station A and K.

Using a one-way ANOVA on ranks (Table 3) to compare the difference between station A and K in the summer periods and winter indicated that these stations were significantly different from one another ($p < 0.001$). Station A was significantly cooler than station K, not just occasionally, but perpetually in the tested seasons.

Figure 4 shows a side-by-side comparison between logger B and K's measured minimum (orange), maximum (dark blue), and average water temperature (yellow) through the entire measuring period, including average air temperature (green–50% transparent) and rain events (light blue columns). Figure 3 supports Figure 4 in that through the measuring period the temperature was higher for the downstream station K compared to the upstream station B. This was especially clear in the period May to June and September to October 2020. In May–June the air temperatures and minimum temperatures were almost the same for station K, while minimum water temperature was 1–2 °C lower than the air temperatures. The downstream K logger also seemed to have higher temperature fluctuations compared to the upstream station B. This was especially clear in December 2020 in which the temperature reached 13 °C at station K, while station B only had a maximum temperature of 7 °C. In February 2021, station B had a longer period in which the temperature was ca. 0 °C compared to station K. This comparison indicates that the stream temperature, both upstream and downstream, were strongly affected by the air temperature. It should also be noted that, as of writing, an unspecified event occurred in March and April 2020 that raised the temperature at the station K stretch of the stream to 20–30 °C. It is yet unknown why the temperature rose so dramatically, but the prevailing hypothesis is that warm water was led out into the stream. This event will not be noted further in this paper, as it was considered to be an unnatural occurrence.

**Table 3.** The average temperature (±SD) of the months January to December (2020–2021) for the stations A–L.

| Month | | A | B | C | D | E | F | G | H | I | J | K | L |
|---|---|---|---|---|---|---|---|---|---|---|---|---|---|
| Jan. | avg ± SD | 3.1 ± 0.9 | 3.2 ± 0.9 | 3.3 ± 0.8 | 3.1 ± 0.8 | 3.3 ± 0.8 | 3.1 ± 0.9 | 3.0 ± 0.9 | 3.5 ± 0.9 | 3.4 ± 0.9 | 3.4 ± 0.9 | 3.7 ± 0.9 | 3.9 ± 0.8 |
| Feb. | avg ± SD | 2.1 ± 2.3 | 2.2 ± 2.4 | 2.4 ± 2.2 | 2.2 ± 2.3 | 2.4 ± 2.3 | 2.2 ± 2.3 | 2.2 ± 2.4 | 2.4 ± 2.4 | 2.4 ± 2.4 | 2.4 ± 2.4 | 2.6 ± 2.2 | 2.9 ± 2.2 |
| March | avg ± SD | 5.3 ± 1.8 | 5.6 ± 1.7 | 5.5 ± 1.6 | 5.4 ± 1.7 | 5.7 ± 1.7 | 5.4 ± 1.7 | 5.4 ± 1.7 | 5.6 ± 1.7 | 5.6 ± 1.8 | 5.6 ± 1.8 | 5.6 ± 2.5 | 5.9 ± 1.5 |
| April | avg ± SD | 8.2 ± 2.7 | 8.3 ± 2.7 | 8.2 ± 2.6 | 8.1 ± 2.7 | 8.3 ± 2.6 | 8.1 ± 2.6 | 8.1 ± 2.8 | 8.3 ± 2.7 | 8.4 ± 2.7 | 8.4 ± 2.7 | 8.6 ± 3.1 | 8.7 ± 2.3 |
| May | avg ± SD | 11.4 ± 2.1 | 11.6 ± 2.2 | 11.5 ± 2.3 | 11.5 ± 2.3 | 11.7 ± 2.3 | 11.5 ± 2.3 | 11.4 ± 2.3 | 11.6 ± 2.3 | 11.7 ± 2.4 | 11.7 ± 2.4 | 11.9 ± 2.6 | 11.8 ± 2.0 |
| June. | avg ± SD | 15.5 ± 1.8 | 15.7 ± 2.0 | 15.9 ± 1.9 | 15.9 ± 2.1 | 16.0 ± 2.0 | 16.1 ± 2.2 | 16.0 ± 2.3 | 16.2 ± 2.3 | 16.5 ± 2.5 | 16.5 ± 2.4 | 16.8 ± 2.3 | 16.3 ± 2.1 |
| July. | avg ± SD | 15.4 ± 1.9 | 15.5 ± 1.8 | 16.0 ± 1.9 | 16.0 ± 2.0 | 16.1 ± 1.9 | 16.5 ± 2.2 | 16.4 ± 2.3 | 16.7 ± 2.3 | 16.9 ± 2.3 | 16.9 ± 2.2 | 17.4 ± 2.1 | 16.8 ± 2.0 |
| Aug. | avg ± SD | 15.2 ± 1.5 | 16.0 ± 2.1 | 15.9 ± 1.6 | 16.3 ± 1.9 | 16.4 ± 1.8 | 16.3 ± 2.1 | 15.9 ± 1.7 | 16.9 ± 2.6 | 16.8 ± 2.3 | 16.7 ± 1.9 | 17.5 ± 2.0 | 16.7 ± 1.6 |
| Sept. | avg ± SD | 14.2 ± 1.5 | 13.6 ± 1.7 | 13.7 ± 1.6 | 13.6 ± 1.7 | 13.8 ± 1.5 | 13.8 ± 1.9 | 14.6 ± 1.6 | 14.2 ± 1.7 | 14.2 ± 1.7 | 14.2 ± 1.2 | 15.4 ± 1.3 | 14.6 ± 1.4 |
| Oct. | avg ± SD | ≠ | 10.5 ± 2.0 | 10.7 ± 1.9 | 10.5 ± 2.0 | 10.7 ± 2.0 | 10.6 ± 2.0 | ≠ | 10.8 ± 2.0 | 10.8 ± 2.0 | 11.0 ± 2.0 | 11.9 ± 1.8 | 11.7 ± 1.7 |
| Nov. | avg ± SD | 4.6 ± 1.2 | 7.8 ± 2.4 | 8.0 ± 2.3 | 6.9 ± 3.3 | 8.0 ± 2.4 | 7.8 ± 2.4 | ≠ | 8.0 ± 2.4 | 7.9 ± 2.5 | 8.4 ± 2.1 | 8.8 ± 2.1 | 8.9 ± 2.0 |
| Dec. | avg ± SD | 4.8 ± 1.2 | 4.8 ± 1.2 | 4.9 ± 1.2 | 4.3 ± 1.5 | 4.9 ± 1.2 | 4.7 ± 1.6 | ≠ | 4.9 ± 1.3 | 4.8 ± 1.3 | 5.1 ± 1.1 | 5.6 ± 1.2 | 6.0 ± 0.9 |

Temperature scale [°C]:

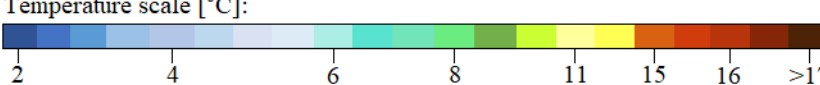

2    4    6    8    11  15  16  >17

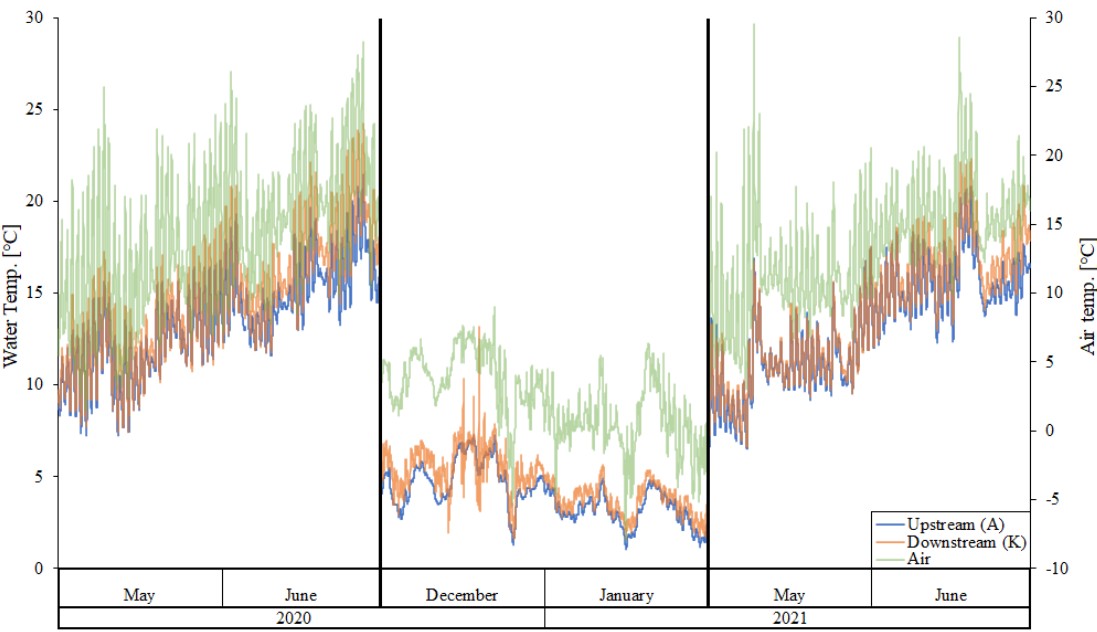

**Figure 3.** Water temperature [°C] (primary *y*-axis) and air temperature [°C] (secondary *y*-axis) shown for the months May–June 2020, December (2020)–January (2021), and May–June 2021 for the upstream station A (blue), downstream station K (orange), and air temperature (green).

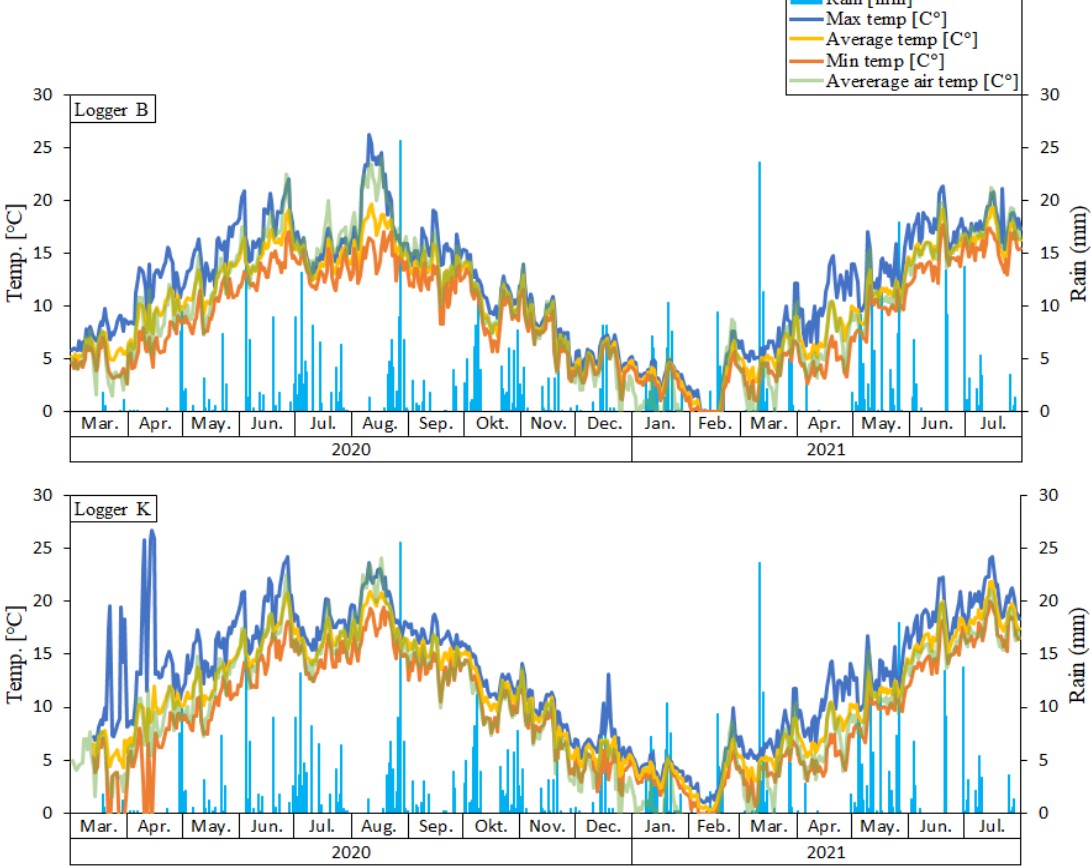

**Figure 4.** Variations in water temperature [°C] presented as the average (yellow), minimum (orange), and maximum (dark blue) temperature, rain [mm] (light blue), and average air temperature [°C] (green—50% transparent) per day at station B and station K.

Figure 5 shows the temperature differences between the maximum and minimum temperatures in Vindinge stream. In late spring, summer, and winter months, the difference in temperatures was often slightly higher at station K compared to station B. The most notable exception was in August 2020, which hints at yet another important factor in heating streams: the water depth. The water depth varied in Stream Vindinge with years and seasons.

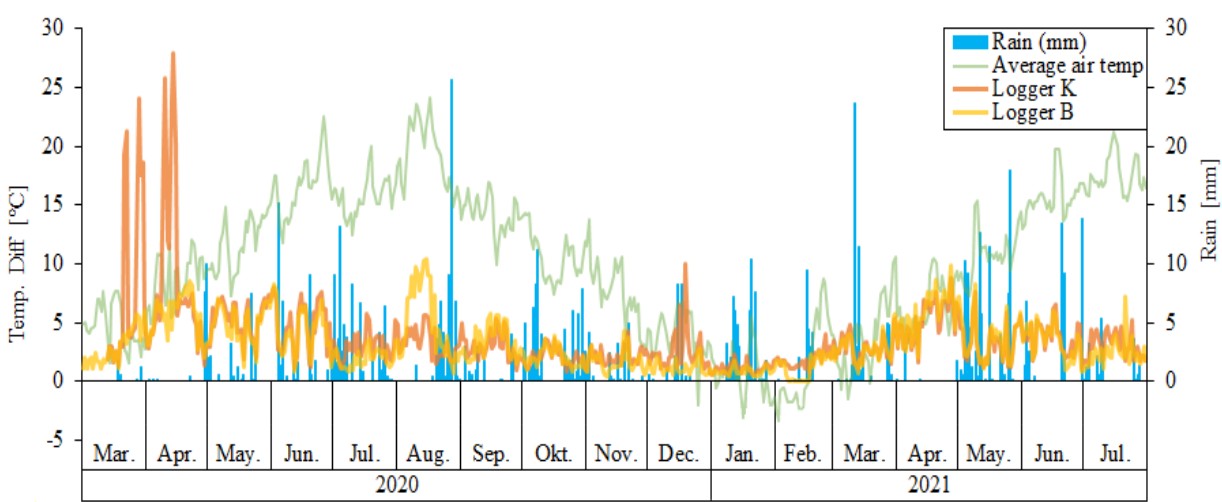

**Figure 5.** Difference between maximum and minimum water temperature [°C], rain [mm], and an average air temperature [°C] per day at station B (yellow) and station K (orange).

Focusing again on station B and K, station K was, on average, the deeper of the two. For 2020 and 2021 collectively, station B had an average depth of 15.1 cm (min = 1 cm, max = 39.4), while station K was roughly 5 cm deeper with an average depth of 20.2 cm (min = 0 cm, max = 48.8). Looking at the seasons (spring and summer collectively for 2020 and 2021 and only 2020 for fall and winter), the largest difference in depth between B and K was in the summer and spring. Station B had an average water depth of 17.3 cm (min = 3.8 cm, max = 38.7 cm) in spring, compared to 11.1 cm (min = 1.0 cm, max = 26.4 cm) in the summer. However, station K had a higher average water depth of 23.9 cm (min = 6.5 cm, max = 45.5 cm) in spring and 18.5 cm (min = 0 cm, max = 45.1 cm) in summer. Water depths in the winter and fall had similar averages (±ca. 2 cm), but very different minimum and maximum depths. In the fall, the average depth for station B was 13.6 cm with minimum 1.9 cm and maximum 25.9 cm. Station K had a similar average depth of 15.4 cm, but a higher minimum and maximum of 6.3 cm. and 36.8 cm. In the winter, station B had an average of 20.5 cm with minimum 10.2 cm and maximum 39.4 cm, and station K had a similar average depth of 22.1 cm, with a lower minimum and higher maximum of 1.5 cm and 48.8 cm, respectively. Figure 5 showed a temperature difference for August 2020 to 10 °C for station B compared to 5 °C difference for station K. In this period station B had a water depth varying from 2–22 cm, contrary to station K with 5–39 cm. This difference in temperature and range in water depth is most likely a result of the drought that hit Denmark in the summer of 2020. Testing the logarithmic correlation of the stream temperature dependency on water level for all seasons and all years, the strongest correlation was seen for station B ($R^2$ = 0.33), while station K showed a weak correlation ($R^2$ = 0.05). By testing the seasons and years separately, it was discovered that the above-mentioned correlation for station B covered over a strong correlation for a lower water level being accompanied by higher temperatures and vice versa, especially in the spring. In the spring, this correlation between lower water depth with increasing temperature was most prominent for station B with an $R^2$ of 0.42 (2020 and 2021 collectively), but was very weak for station K with an $R^2$ of 0.10. All correlations had a *p*-value < 0.0001.

### 3.2. The Effect of Different Outlet Types

In Figure 6, the average seasonal temperatures for the different stations with the different outlet types are illustrated. As expected, the temperatures increased seasonally in this order: winter (3.2–4.0 °C) < spring (8.8–9.9 °C) < fall (4.6–12.0 °C) << summer (14.9–17.2 °C). The only exception was at station A, in which spring had higher temperatures than fall. The temperatures in spring and fall did not vary much between stations or distance, but some variations were seen in summer and winter. As previous figures have indicated, the temperature increases with distance through the stream. A one-way ANOVA analysis between the different outlet types in the different seasons proved that there were no statistically significant differences between the outlet types in any season.

### 3.3. Different Outlet Types of Response to Rain Events

Figure 7 shows examples of the four different outlet types (C: rain + basin, E & H: Common overflow and I: rain–basin), and the temperature responses to rain events in May–June 2020 are shown. In May the outlet temperature follows the same overall temperature trend, but outlet I ($\div$basin) consistently had the highest temperatures, up to 0.75 °C warmer that the other outlets. Outlet I ($\div$basin) was followed by the common outflow outlets E and H and lastly outlet C (+basin). However, after the 8th of June, the temperature differences became more pronounced. Following the rain event on the 18th, outlet I ($\div$basin) was almost 1.5 °C warmer than outlet C and E. This temperature difference kept rising until it topped on the 27th, where outlet I > 2.0 °C compared to C and E. However, there were no statistical differences between the outlets as $p > 0.05$ between all outlet types (ANOVA-Tukey test). Observing the effect of the different rain events closely, it can be concluded that whether the stream temperature will rise or fall with the rain event will change from event to event.

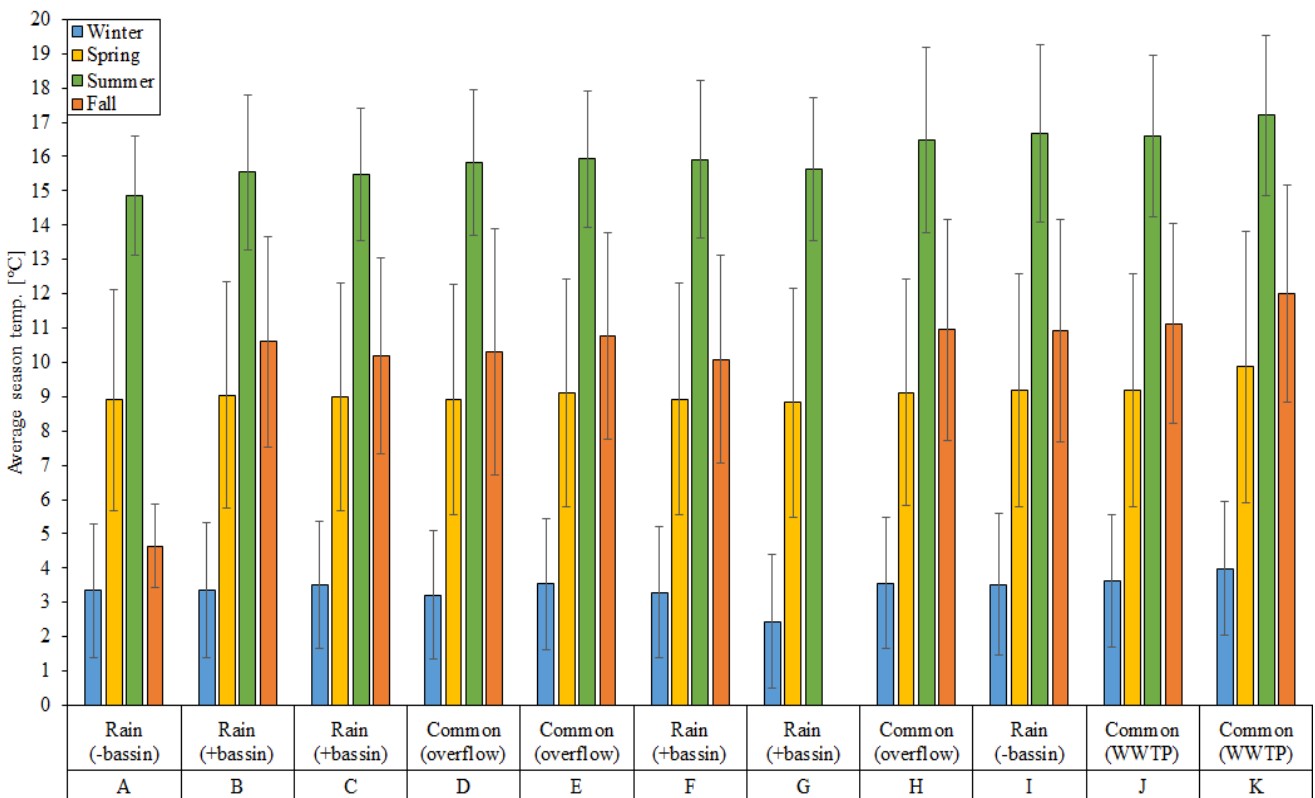

**Figure 6.** Average water temperature at each station (A–K) pr. season: winter (blue), spring (yellow), summer (green), and fall (orange). Each station is listed with the nearest upstream outlet type: rain (-basin), rain (+basin), and common (overflow and WWTP).

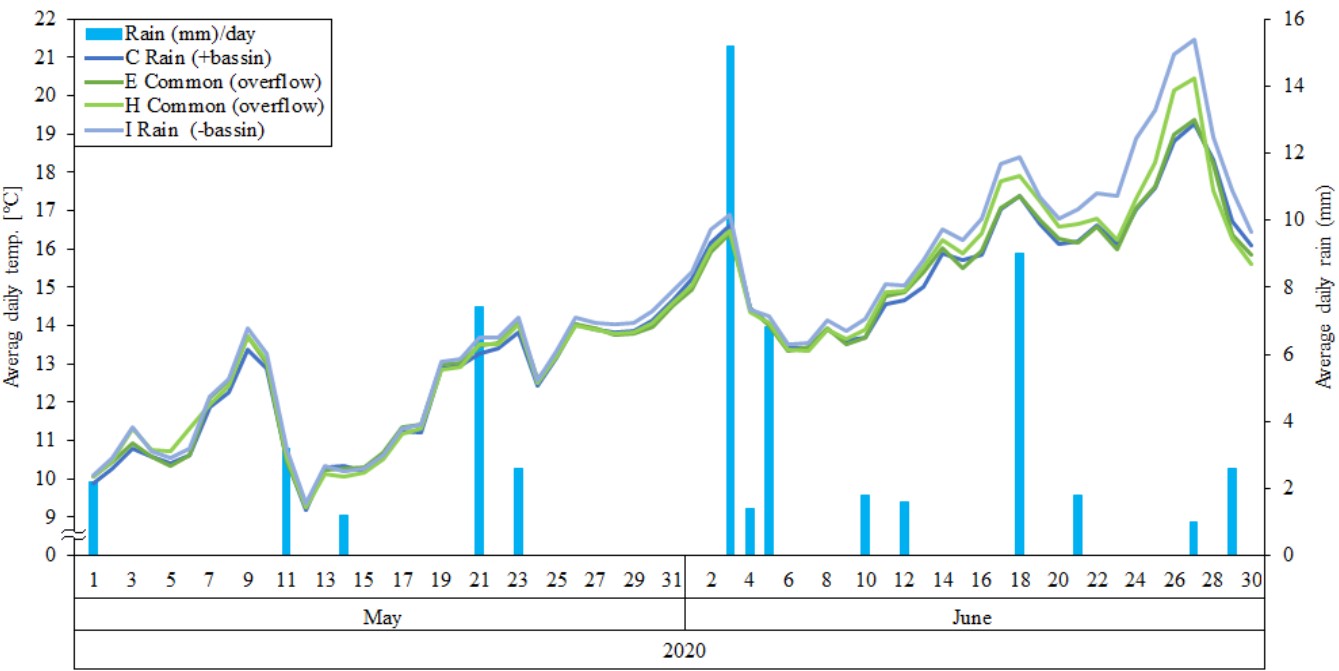

**Figure 7.** Variations in stream temperature [°C] downstream of specific outlet types during rain events [mm] (C: rain + basin, E & H: Common overflow and I: rain–basin). From the period May–June 2020.

## 4. Discussion

### 4.1. Urbanization Effect on the Stream Temperature

It has long been a known fact that recipients of urban watersheds will be affected by the characteristics of the catchment area. However, it has also been assumed that these affects are mostly in connection to stormwater events [8,43,46]. In this study, we have shown that this might not be the case. An urban landscape might affect and aid the heating of the recipient streams not only during rain events, but continuously through the year with seasonal variations.

All graphs in this paper have indicated that there was a close relationship between the air temperature and the water temperature for all stations, and also that the urban areas surrounding the stream have an UHI effect. This is no surprise, as others have shown similar relationships in the past [51–54]. What has not been published previously is the collective impact of the village on the stream temperature. In Figure 3 and Table 3 it is shown that there was a significant difference between the temperature measured at the upstream station A and the downstream station K throughout the summer and winter of 2020–2021. This difference was not only during rain events, but also in dry weather periods, indicating that there is a constant heating effect inflicted by the town on the stream. Figures 4 and 5 further highlight this tendency.

### 4.2. Temperature Rise through Aarslev Village

Several of the figures and statistical analyses in this paper all point to the same conclusion: the temperature in Stream Vindinge rises as the stream progresses further through Aarslev village, and the upstream and downstream stations can have very different responses to specific events. During summer 2020 there was a drought in Denmark. This made the correlation between water depth and temperature stronger in the spring. Figure 5 and accompanied statistics showed that the upstream station B was more affected than the downstream station K by the drought. This was expressed by a large difference in temperature and a low water depth. Under these circumstances, the downstream station kept a more constant temperature and had a higher depth. From the former results, it would have been expected that the downstream station K would have been more affected by the

drought than the upstream station. There are some explanations for why this might not be the case. One is that there is very little shelter and shade before the town, but some shade is provided through the village, both from vegetation and constructions. Additionally, some water might be supplied by the stream's collective catchment area, both from natural infiltration and occasionally from multiple outlets.

### 4.3. Temperature Differences Affected by Outlet Types

From Figures 6 and 7 and accompanied ANOVA analysis, it was concluded that there was no overall difference between the temperatures in relation to the different outlet types. This was an unexpected result. A possible explanation as to why is the overall placement of outlets relative to each other. The outlets are very spread out through the village, and the overall heating from the village might neglect any individual differences between outlet types. It was expected that common overflow outlets would be warmer than rainwater outlets and further expected that the temperature would increase downstream stormwater ponds outlets compared to direct rainwater outlets, especially in relation to rain events.

The was due to previous studies having indicated that water discharged from stormwater ponds can be warmer than the incoming water to the ponds during rain events [25,55,56]. It has been suspected that the reason for this heat increase is a combination of insufficient shading and incoming solar radiation heating up the still water [25]. Jones et al. [25] conducted a study in North Carolina, USA, in which they tried to determine the effect of stormwater wetlands and wet ponds on the temperature of urban runoff. They measured stormwater runoff to have an average temperature of 24.70–26.55 °C in June–August 2006 before entering the pond. However, when the stormwater reached the pond, the temperature had, on average, decreased. The pond received water from two inlet types: one metal and one concrete. The water that travelled through the metal pipe had an average effluent temperature of 18.66–21.71 °C to the pond, while the water from the concrete pipe had temperatures of 22.19–24.61 °C. The effluent temperatures from the pond into the stream, however, were 24.94–27.74 °C, meaning the temperatures rose with 1.57–8.98 °C. It was concluded that this difference in influent and effluent temperatures indicated that the pond was a source of thermal pollution. It was discussed that this might be mitigated with further shading and possibly new outlet structures that drain the pond from the bottom and not the surface, as the temperature was lower at 120 cm depth compared to 40 cm depth. Sabouri et al. [57] made a similar conclusion in their examination of thermal impact from stormwater ponds. Sabouri et al. [57] examined six stormwater ponds in Ontario, Canada, and showed an increase in event mean temperature of 1.1 °C to 7.0 °C between the inlet and outlet. Sabouri et al. [57], like Jones et al. [25], concluded that water drained from the bottom, at 90–120 cm depth, was cooler than water drawn from above this depth. Stormwater management in Denmark is already designed with several of these design guides: water is let to the ponds and streams in underground pipes and the pond outlets are often submersed. Shading varies with location. The ponds in Aarslev were also designed following these guidelines. The direct outlet of stormwater to the streams might be further cooled by extending the length of the underground pipes. The ratio between the pond's wet volume and impermeable area in the catchment [m$^3$/red.ha] and thus the residence time is likely also a factor to be considered when speculating how much, or how fast, the influent stormwater can be heated. From Figure 7, it was observed that the stream temperature can both increase and decrease in relation to a rain event. It was speculated that this was due to a "first heat flush" with subsequent cooling. As previously mentioned when paved surfaces are heated, this heat dissipates to the cooler stormwater during rain events [10,11]. Figure 7 indicates that if the rain event is of a high enough intensity, the stream will be cooled by the rain event after the "first heat flush". Previous studies by Martin et al. [58], Jones et al. [59], and Hørup et al. [26] similarly showed that thermal pollution has a peak in the initial period of a rain event, similar to what has been confirmed to occur for other pollutants, commonly referred to as "first flush" [60–63].

Hørup et al. [26] studied how a stream recipient was affected by outlet from a stormwater pond in Voldum in Eastjutland. They found that the temperature effect of the stormwater pond was not as severe as described here. The largest temperature change in the receiving Revens Mollebak, attributed to the stormwater pond, was 1.5 °C, and only for a short period of time during rain events. At all other times the temperature impact was miniscule. Hørup et al. [26] concluded that since the pond outlet was placed 3100 m downstream from the spring and received water from many different outlets (drains, direct rain outlet, other ponds, etc.), the examined stormwater ponds discharge was miniscule compared to the total waterflow. It was, however, speculated that the pond would have a greater impact on the stream had it been placed closer to the spring.

Predicting whether a rain event will heat or cool the stream, or how long it will last, might not be as straightforward to determine. The cooling or heating effect is likely a combination of factors such as the number of sunlight hours, air temperature, and days since last rain event, overcast or clear skies, wind intensity, catchments characteristics, etc. Determining such a factor would require further extensive research of different streams and stretches as it would likely vary with location, catchments characteristics, and climate.

### 4.4. Ecological Condition and Consequences for Trout and Macroinvertebrates

The average summer temperature at station K was 17.24 °C, with some instances of the stream temperature rising to 25 °C at station B in August 2020 (Figure 4). According to instructions from the Danish Environmental Protection Agency, a stream's ecological condition can, among other factors, be determined based on the stream's temperature (°C). The ecological condition categories are high, good, and moderate [64]. In streams with a high and good ecological condition, the summer temperatures should not exceed 21.5 °C and 25 °C for moderate streams. The maximum change in temperature after an outlet should not exceed 1 °C for high and good, while >3 °C signifies a moderate stream. For both station B and K, the max summer temperature recorded was ca. 26 °C, and the temperature difference between these stations reached 1.9 °C in the summer. Based on these recorded temperatures, and if the entire village was seen as a point source, this stretch of Stream Vindinge has a good, but leaning on moderate, ecological condition. This conclusion would be expected to have some impact on Vindinge streams' ecosystem.

For adult trout the CTM of 30 °C [32] was not exceeded, and for alevins, paars, and smolts, the CTM were only occasionally exceeded, but on average the temperature was well below their CTMs of 20–30 °C. Young fish undergo the physiologically changes into smolts in the spring, where the average temperature measured was 8.94 °C, well below their CTM of 22–30 °C [33–35]. Maximum temperature in the examined area rarely exceeded 20 °C, but from June–August there were periods where this temperature was exceeded, more often at station K than B. These temperatures were similar to what was measured in the North Carolina study [25], in which the effluent temperatures from the wet pond did not drop below 21°C from June to August, with a maximum temperature of 29.2 °C. It was concluded that these elevated temperatures might have a negative effect on the trout population [25]. The maximum temperatures measured in Vindinge stream were lower than the ones found in North Carolina. However, the max temperature measured in Stream Vindinge might be slightly high for alevins.

The mean UTT for several species of macroinvertebrates was determined, and for stoneflies, caddisflies, and mayflies, this temperature limit lies in the range of 22.3–30.1 °C. However, the summer temperatures might have negatively impacted stonefly-nymphs, as they thrive in temperatures from 12–20 °C [36]. It is unlikely that the registered temperatures would kill either trout or invertebrates, but the ecosystem can, as the determined good–moderate ecological condition also implies, be severely affected before thermal stress would be deadly to living organisms.

Aside from thermal stress, higher water temperatures also mean a lower $O_2$-dilution and an increased bacterial activity, both of which will lower the $O_2$-concentrations. This might be very damaging for both oxygen and temperature sensitive species such as stone-

flies, caddisflies, and mayflies. These animals, which indicate a healthy and thriving ecosystem, use skin respiration and thus depend on having low water temperatures and high oxygen concentrations. A decline in invertebrate population will also likely reinforce the effect on the trout, as freshwater invertebrates are the trout's food base.

As temperatures for both lakes and streams are projected to rise in the future, these negative effects would not be expected to become less severe or frequent [65–67]. European river temperatures are projected to increase with 1.6–2.1 °C 2071–2100 compared to the recorded river temperatures in 1971–2000 [66]. According to a study from O'Reilly et al. [67], Northern European lake surface summer temperatures are expected to rise with between 0.36–0.72 °C/decade. This expected rise in temperature will likely also be the case for stormwater ponds, further intensifying the previously discussed "first heat flush". This might be problematic for the entire freshwater ecosystem, as it will affect all trophic levels.

According to Till et al. [68], the rise in northern temperate climates freshwater temperatures can lead to more frequent events of fish die-offs as a direct result. These events are projected to become more frequent as the yearly average temperature rises above 3 °C, which is projected to happen within the next 80 years [68]. A study on fish adaption to climate change and rising temperature from Crozier et al. [69] concluded that rising temperatures will affect timing of migration and reproduction, age at maturity, age at juvenile migration, growth, survival, and fecundity, but also that many species will likely be able to adapt to these new circumstances.

### 4.5. Decrease in Temperature after Urbanized Area

It is expected that a stream's temperature will rise the further from the spring the stream travels due to the air temperature and sunlight hours [70]. However, the temperature increase demonstrated in this paper is too high to be a completely natural process over a 5.2 km distance. Exactly what the natural temperature of Vindinge stream would be in this area is unknown, but it does raise the following question: will the stream temperature decrease after the city? The answer seems to depend on the season. Logger L was purposely placed 1650 m after Aarslev village at a point where it was suspected that the temperature might have decreased due to the lack of inputs/outlets. In the months November–April, station L was warmer than station K (0.1–0.4 °C), but in May–October, station L was roughly 0.1–0.8 °C warmer than station K. Whether or not the temperature would further decrease downstream would be a topic for future investigations.

### 5. Conclusions

In this paper, it was concluded that urbanization in the form of the rather small Aarslev village does have a heating effect on Stream Vindinge that runs through the village. This effect was not only in connection to rain events, as has previously been assumed, but continuously through the seasons, most notably in the summer. Future studies should be focused on determining if other streams in urban areas, villages, towns, and cities of greater and smaller sizes are similarly affected. No significant difference between the outlet types could be determined. It was speculated that this might be due to the overall placement of the outlets in the stream and the overall heating effect of the village might negate individual differences between the outlet types. It was expected that overflow > rainwater outlets (with pond > direct), due to previous studies having made such conclusions. It might also be because Danish outlet design corresponds with recommendations from previous studies. These include submersed outlets from ponds and leading stormwater to streams and ponds through underground pipes. Rain events could not be concluded to uniformly heat or cool the stream, but a "first heat flush" was seen with subsequent cooling. Future studies on the "first heat flush" should be dedicated to measuring the transference of heat energy from surfaces to runoff. An example could be to measure surface temperature changes before and after a rain event and the corresponding changes in recipient stream temperature.

The intensity of heating or cooling was speculated to differ with location, diurnal cycle, air temperature, days since last rain event, weather conditions, catchments characteristics, and outlet design. Future research should be dedicated to determining which factors are most influential on the "first heat flush" effect. As this is likely to vary with location and climate, it would need to be extensively examined in different regions. According to instructions from the Danish Environmental Protection Agency, the stretch of stream going through Aarslev village is between a good–moderate ecological condition. This would be expected to impact the fish and macroinvertebrates communities. The measured temperatures would likely not lead to death due to thermal stress, however accompanied lower $O_2$-concentrations due to higher mineralization and less dissolved oxygen could be detrimental to the ecosystem. Lastly, it was concluded that the stream temperature did decrease 1650 m after Aarslev village.

The study could have been improved by monitoring additional streams running through similar villages. Furthermore, a more direct comparative element could have been added in the form of monitoring a similar stream stretch without a village. Expanding the period of data collection to more than 2 years might also have been beneficial.

**Author Contributions:** Conceptualization, A.S.K. and S.E.; methodology, A.S.K. and S.E.; software, A.S.K.; validation, A.S.K. and S.E.; formal analysis, A.S.K.; investigation, A.S.K.; resources, A.S.K. and S.E.; data curation, A.S.K.; writing—original draft preparation, A.S.K.; writing—review and editing, A.S.K. and S.E.; visualization, A.S.K.; supervision, S.E.; project administration, S.E.; funding acquisition, S.E. All authors have read and agreed to the published version of the manuscript.

**Funding:** This research received no external funding.

**Institutional Review Board Statement:** Not applicable.

**Informed Consent Statement:** Not applicable.

**Data Availability Statement:** The data presented in this study are available on request from the corresponding author.

**Acknowledgments:** We thank Faaborg-Midtfyn Municipality who owns and manages the examined part of Vindinge stream for providing us with background data and information throughout the study period making it possible for us to perform this study. We also thank the lab technicians Rikke O. Holm, Janus T. Jensen, students Mads E.D. Greisen and Andreas H. Hansen at the University of Southern Denmark for helping with field work and data collection.

**Conflicts of Interest:** The authors declare no conflict of interest.

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
