# Peer review of "Influences of Urban Discharges and Urban Heat Effects on Stream Temperature"

_hydrology, doi:10.3390/hydrology10020030_

Round 1
Reviewer 1 Report
The manuscript has an interesting idea, and it is in the scope of the journal. It is well organized and presented. However, there are several comments need to be clarified as shown below. The reviewer recommend that this paper can be published in this journal after a minor revisions.
• The methodology part doesn't clear; it may not catch the attraction. Try to make it more concise and briefer.
• The introduction in the paper is very short need to extend
• Why the author considered these data (2020-2021) however that date now (2023)?
- I recommend combining section results and discussion into one section.
- Table 2 must correct to Table 3 and Table 3 to Table 2
• There are a lot of typos related to the superscript in the text
Author Response
Thank you very much for your time and comments. I have tried to comply with your recommendations as follows:
- The methodology part doesn't clear; it may not catch the attraction. Try to make it more concise and briefer.
- Sections of the method chapter has been rewritten and condensed.
- The introduction in the paper is very short need to extend
- Yes, it was very short and needed to be extended. The introduction has been extended with topics about:
- Climate change projected changes to heat and precipitation
- Expanded on UHI
- Expanded on wet stormwater ponds
- Yes, it was very short and needed to be extended. The introduction has been extended with topics about:
- I recommend combining section results and discussion into one section.
- I have elected not to combine the result and discussion section as this would be in conflict with the template provided by Hydrology.
- Table 2 must correct to Table 3 and Table 3 to Table 2
- This has been done.
- There are a lot of typos related to the superscript in the text
- The manuscript has been re-read and these amended.
- Why the author considered these data (2020-2021) however that date now (2023)?
- Due to personal reasons, it could not be published sooner.
Once again thank you very much for your time and effort.
Reviewer 2 Report
The article is interesting and shows the important issue of human impacts on aquatic ecosystems. The authors are based on detailed, albeit brief, field observations of stream Vindinge. The layout of the article is logical, the statistical methods used and the graphical part adequate. I recommend the article for publication after taking into account the following comments.
I disagree with the statement: "However, urbanizations impact on stream temperature is a less studied subject especially regarding seasonal changes". Please expand the literature review in this regard. How does the presented research fit into the existing state of knowledge?
The essence of the article is to demonstrate the impact of urbanized areas on stream thermal conditions. However, as shown in Figure 1, the stream is not in contact along its entire length with this type of use. A large part of the stream (section E-H) is adjacent to agricultural land on one side. What effect does the growing season (amount of water retention) have on temperature changes? Was the same vegetation present in the two seasons analyzed?
In the discussion section (as well as in the abstract), the authors refer to the lake study. Given the different nature of the two environments (rivers/lakes) and especially the different thermal characteristics (including stratification), such an approach is inappropriate. There are many studies in Europe aimed at water temperature changes in rivers. Please amend this part of the paper and also the reference in the abstract.
Author Response
Thank you very much for your time and your comments. I have tried to comply with your recommendations as follows:
- I disagree with the statement: "However, urbanizations impact on stream temperature is a less studied subject especially regarding seasonal changes". Please expand the literature review in this regard. How does the presented research fit into the existing state of knowledge?
- This statement was wrong and did not properly express the intent with the study. It has been rewritten. Thank you for catching it.
- A large part of the stream (section E-H) is adjacent to agricultural land on one side.
- This landscape is considered urban in Denmark. With 61% of the land being used for agriculture, it is almost impossible to find any stream that has no agricultural area in either catchment or vicinity. However, the fields present here does not drain to Vindinge stream. The ground in this area is mainly sand and gravel (gravel excavation pits are present outside the village), thus draining the fields to the stream is not necessary. Some diffuse runoff is likely present, but because of the soil composition we are very comfortable that it is the village that is exerting most of the stress on the stream. It is possible that the diffuse runoff could be affected in the growing season as a higher water uptake would reduce the diffuse runoff.
- Was the same vegetation present in the two seasons analyzed?
- Yes it was the same crops each year.
- In the discussion section (as well as in the abstract), the authors refer to the lake study. Given the different nature of the two environments (rivers/lakes) and especially the different thermal characteristics (including stratification), such an approach is inappropriate. There are many studies in Europe aimed at water temperature changes in rivers. Please amend this part of the paper and also the reference in the abstract.
- This part was ill explained and should not have been included in the abstract as it was. The part in the discussion about the lakes projected temperature increase was not supposed to be equated to the temperature effect on the streams directly, but to the effect of the stormwater ponds and through them the stream. However, this was poorly explained and has been rewritten and expanded with the projected temperature increase of streams. Appropriate changes were also made to the abstract.
Once again thank you very much for your time and effort.
Reviewer 3 Report
1 General Remarks
Recommendation: Minor Revision
Summary: In this manuscript, the authors attempted to examine the temperature effect of the urban village Aarslev on stream Vindinge in Southern Denmark. Specifically, a stretch of stream Vindinge that passes through Aarslev village in Southern Denmark was monitored using stationary temperature and pressure loggers for two years. The authors investigated whether this urban landscape has a heating effect on the receiving Stream Vindinge. The topic well falls within the scope of Hydrology. It is essential to understand the urbanization impacts on stream temperature because an increase in stream temperature can directly affect the animals that live in streams. Overall, the authors have conducted many experiments with good experiential designs. Meanwhile, the results were well demonstrated in this manuscript with an in-depth discussion. I suggest accepting the publication of this manuscript in its present form except for one minor revision.
2 Comments
1. The Conclusion section of a manuscript should include a summary of the results, the potential limitation of this research, and the recommendation of future research directions for peers. However, only a summary of the results was covered in the Conclusion section of this manuscript. It would be great to add the potential limitation of this research and the recommendation of future research directions for peers.

Author Response
Thank you very much for your time and your comments. I have tried to comply with your recommendations as follows:
- It would be great to add the potential limitation of this research and the recommendation of future research directions for peers.
- You are right - the potential limitation and future recommendations of research directions should have been included to begin with and is now included.